# OpenReview forum: "Retention Score: Quantifying Jailbreak Risks for Vision Language Models"
_NeurIPS.cc/2024/Workshop/SafeGenAi — SafeGenAi Poster_

### Official Review · Reviewer_bNxC · 2024-10-09
**good work, might need more exaplanations**

**Rating:** 7
**Confidence:** 3

**Review:**

##  Summary
The authors propose a Retention Score to evaluate jailbreak risks in vision-language models. The method leverages diffusion models to generate additional samples and measure the margin of toxicity in order to assess robustness. This approach allows for a faster evaluation compared to traditional adversarial attacks.

### Strengths:
- The topic is highly relevant and important given the widespread use of large language models today.
- The authors provide a detailed explanation of their experiments.
- The work explores both image and text attacks, expanding the scope of robustness evaluation.

### Weaknesses:
- Some aspects of the writing and presentation are difficult to follow. For example, the terms "Retention-I" and "Retention-T" are mentioned multiple times before being explained as "Retention-Image" and "Retention-Text," respectively. Additionally, although time efficiency is emphasized, there is only a brief section discussing it, and the figure illustrating this is very small.
- The paper does not address an important assumption: whether the model is already inherently toxic or non-toxic. If the model is toxic, it may produce harmful responses regardless of an adversarial attack. If this is true, how can the ASR be used to demonstrate robustness?- - The results of the Retention Score require further discussion. For instance, in Table 1, the LLaVA model's "man" group has a Retention Score of 0.1917, corresponding to an ASR of 52.16%, while the InstructBLIP model's "old" group has a score of 0.4861, with a similar ASR of 52%. Despite the similarity in ASR, the Retention Scores are markedly different. This discrepancy raises questions about the consistency of the Retention Score in indicating risk. If the Retention Score is not aligned with actual risks, how can this metric be considered reliable or explainable?

---

### Official Review · Reviewer_JYTv · 2024-10-10
**Generally good idea, but...**

**Rating:** 7
**Confidence:** 4

**Review:**

The authors introduce a novel multimodal framework, *RetentionScore*, which provides a conditional robustness certificate against jailbreak attempts from both visual and textual inputs. While the motivation is clear and the framework is well-explained, there is a notable issue: the claim that optimization-based attacks are impractical is not sufficiently backed by evidence. The authors fail to specify how much time it would take to develop an adaptive attack tailored to their method. Moreover, the pipeline's reliance on two diffusion-based models before inputting the modalities into the VLM, along with the use of a 70B model and the "Perspective-API" for calculating the RetentionScore, makes the process seem computationally intensive. How long does it actually take to compute RetentionScores and generate robustness certificates? Additionally, the runtimes provided in Table 7 lack context—how many samples were tested to arrive at these figures? Clarifying these aspects would significantly improve the strength of the paper.